# Comparison of Fatty Acid Profile in Egg Yolk from Late-Age Hens Housed in Enriched Cages and in a Free Range System

**DOI:** 10.3390/ani14071099

**Published:** 2024-04-04

**Authors:** Meng Peng, Siria Tavaniello, Mirosław Banaszak, Sebastian Wlaźlak, Marisa Palazzo, Giulia Grassi, Giuseppe Maiorano

**Affiliations:** 1Department of Agricultural, Environmental and Food Sciences, University of Molise, 86100 Campobasso, Italy; m.peng@studenti.unimol.it (M.P.); m.palazzo@unimol.it (M.P.); giulia.grassi@unimol.it (G.G.); maior@unimol.it (G.M.); 2Department of Animal Breeding and Nutrition, Faculty of Animal Breeding and Biology, PBS Bydgoszcz University of Science and Technology, 85-084 Bydgoszcz, Poland; miroslaw.banaszak@pbs.edu.pl (M.B.); sebastian.wlazlak@pbs.edu.pl (S.W.)

**Keywords:** late-age hens, enriched cages, free range, egg yolk, fatty acid profile

## Abstract

**Simple Summary:**

A ready supply of eggs has been provided for consumers throughout the years at a relatively low price sufficient to meet consumer demand. In response to the cage-free trend, adequate studies regarding the difference between the cage system and alternative systems are necessary due to the growing concerns about food safety and animal welfare, especially animal living conditions. Along with welfare concerns, there is rising consumer consciousness about the environmental sustainability of animal production. From this perspective, increasing persistency in lay and stability in egg quality would generate environmental and economic benefits. The present study investigated the effects of the enriched cage and free-range systems on the egg yolk nutrient composition at the late phase of the laying cycle (from the 68th week to the 74th week of age). The results of the present study indicated that the egg yolk lipid profile in terms of the total lipid, cholesterol, and fatty acid composition was not modified by the increasing age of laying hens. On the other hand, the results confirmed that the free-range system may improve the nutritional yolk fatty acid profile and its positive impact on human health, which can be a convincing support for the trend of “cage-free” eggs.

**Abstract:**

In recent years, the free-range system for laying hens has increased, driven by societal sensitivity to animal welfare. This study aimed to comparatively analyze the total lipid, cholesterol, and fatty acid composition of egg yolks of late-age laying hens reared in enriched cages (C) and the free-range system (FR). Eggs were collected from Lohmann Brown Classic hens at the 68th, 70th, 72nd, and 74th week of age. The concentrations of total lipids and cholesterol were not affected (*p* > 0.05) by either factor. Egg yolk from the FR group showed lower (*p* < 0.01) monounsaturated fatty acids and higher (*p* < 0.01) polyunsaturated fatty acid (PUFA) compared with that of the C group. From a nutritional point of view, the PUFA n-6/n-3 and the PUFA/SFA ratios of egg yolk from the FR group were favorably lower and higher (*p* < 0.01) compared with the C one. Conversely, hen age did not affect (*p* > 0.05) the fatty acid composition of yolks. Interactions between factors were found for total n-3 and n-6 PUFA and the n-6/n-3 ratio (*p* < 0.01), as well as the thrombogenic index (*p* < 0.05). In conclusion, the results confirmed that the free-range system may improve the nutritional yolk fatty acid profile and its positive impact on human health.

## 1. Introduction

Eggs are considered an affordable source of essential nutrients because they are rich in amino acids, lipids, vitamins, minerals, and several biologically active compounds. Furthermore, eggs are a means of providing essential and beneficial fatty acids in the human diet. Lipids from eggs are rich in mono-unsaturated fatty acids and can be easily enriched with n-3 polyunsaturated fatty acids, incorporating selected lipid sources into the diet [1,2,3]. However, eggs are also high in cholesterol. Cholesterol was considered in the past to be a negative nutrient, although the risk assessment has evolved in line with more recent evidence suggesting a smaller impact of dietary cholesterol on cardiovascular disease risk than saturated fat [4,5]. The internal and external qualities of eggs can be influenced by various factors, e.g., the hen’s genetic type, laying hen age, feeding strategy, and animal welfare; the latter is linked to the production system [3,6]. In recent years, there has been a growing interest among consumers to better understand the relationship between the production system, animal welfare benefits, and higher-quality egg production [7]. 

In relation to the global increase in the human population, global egg production has increased exponentially thanks to intensive egg production systems, mainly conventional cages [8]. Over the last few decades, alternative production systems (barn housing, free-range, and organic systems) have been introduced, driven by consumer demand for environmentally friendly and healthy foods from animals raised in better rearing conditions and animal welfare [9,10,11,12,13]. Since 2012, in the European Union, conventional cages have been banned, but enriched cages are still allowed, characterized by structural improvements meant to enhance hen welfare. The “End the Cage Age” campaign started in 2018 to support the EU’s ban on the cage system [9]. In a recent scientific opinion on the welfare of laying hens on farms, the European Food Safety Authority (EFSA) recommended that all EU egg production should become cage-free [14]. In Europe, laying hens produce close to 6.8 million tons of eggs each year. Data from the EU market situation for eggs shows that laying hens are kept mainly in cage-free systems (barns: 37.8%, free-range: 15.5%; organic: 7.1%) compared with enriched cages (39.7%) [15]. Obviously, in responding to the cage-free trend, adequate studies regarding the difference between the cage system and alternative systems are rewarding. An abundance of research supports the significant influence of egg quality traits, including egg weight and shape; eggshell color; weight; thickness; density and strength; albumen and yolk traits; and egg nutritional quality, in different production systems [11,12,13,16,17,18,19,20]. However, in a recent systematic review regarding the effects of the hen housing system on egg quality [6], inconsistency was found in the results of the 50 articles considered (from 1992 to 2020), as was a lack of standardization in the methodology of these research studies. The authors suggest that in-depth studies evaluating different housing systems under the same conditions and evaluating the interactions of factors such as genetics and the age of the animals, thus assessing multiple effects on egg quality traits, are warranted.

Along with welfare concerns, there is rising consumer consciousness about the environmental sustainability of animal production. From this perspective, increasing persistency in lay and stability in egg quality to 90–100 weeks would generate environmental benefits. Indeed, Bain et al. [21] estimated that even 25 more eggs per hen could potentially reduce the UK flock, including breeding hens, by 2.5 million birds per year. Considering the actual production rate, producers and farmers could keep eggs until the end of the laying phase. In fact, Karcher et al. [22] reported that hen-day egg production in different housing systems is maintained at a high level of over 85% until the 77th week of age. In light of the above-mentioned aspects regarding hen welfare and the environmental and economic sustainability of egg production, the present study aims to evaluate the effects of the production system (enriched cages vs. free range) and hen age at the late laying phase on the egg yolk nutrient composition.

## 2. Materials and Methods

### 2.1. Ethics

The conditions for keeping hens in both production systems (enriched cages vs. free range) complied with the rules of Council Directive 1999/74/EC. 

### 2.2. Animals and Samples Collection

Table eggs were obtained from a company in Gostków (Poland) that produces eggs in the south of Poland within the Lower Silesian Voivodeship. The eggs were laid by Lohmann Brown Classic laying hens that were kept in two different breeding systems: enriched cages (C) and free range (FR). The hens were kept in accordance with the recommendations of Lohmann Tierzucht GmbH (Am Seedeich, Cuxhaven, Germany), the breeder of laying hens. The air temperature in the hen house was maintained between 18 °C and 20 °C, with a humidity range of 60–70%. The concentration of harmful gases such as CO_2_, NH_3_, and H_2_S did not exceed the permissible standards, i.e., CO_2_—3000 ppm, NH_3_—20 ppm, and H_2_S—5 ppm. From the 21st week, both production systems received 14 h of light with a maximum intensity of 15 lux and 10 h of darkness. The infrastructure of the free-range facility used in the experiment met the welfare requirements; i.e., the density was 2 hens/m^2^. The free-range area was fenced and equipped with shelters and nipple drinkers. The grass sown was a mixture of low meadow grasses. Access to the free range was provided during daylight hours. Natural light was a determinant of the lighting program in the facility. The enclosure was hygienized before the facility was inhabited and secured in case of suspected zoonotic threats. During the research, there was no epizootic threat in the farm area. The birds were fed a commercial diet (Table 1) in accordance with the current nutritional recommendations for poultry, as stated by Smulikowska and Rutkowski [23]. Feed and water were provided ad libitum.

The C system requirements were in line with the recommendations for improved laying hen welfare. The cage area was greater than 2000 cm^2^, providing more than 750 cm^2^ of space per hen. Each cage had a nest, bedding, perches (15 lengths per animal), a feeder (12 cm long per animal), and nipple drinkers. In the FR system, 9 laying hens occupied 1 m^2^ of usable floor area, with a maximum of 28,704 birds in the room. The litter covered at least 1/3 of the floor area, and oblong feed containers (10 cm long per bird) and wheeled feeders (4 cm per bird) were used. Water was supplied through nipple drinkers (10 pieces per bird). Group nests were placed, the area of which was within the permissible limits (120 pieces/m^2^). The open-run area was adapted to the number of hens and the land type. 

A total of 120 (60 from C and 60 from FR) intact, clean, and normal-shaped eggs were collected randomly at the 68th, 70th, 72nd, and 74th weeks of hen age (15 eggs for each time point). Eggs were transferred to the laboratory and stored at 4 °C for 24 h, and the yolk was separated and stored at −20 °C until analyzed. The evaluation was performed on individual eggs.

### 2.3. Total Lipid and Cholesterol Content

In total, 3 g of each individual yolk was mixed with 60 mL of a 2:1 (*v*/*v*) mixture of chloroform and methanol; further procedures were performed according to the method of Folch et al. [24] to extract the total lipids. Lipids were separated from the chloroform phase; dried to constant weight; weighed; and then recovered with 10 mL of mixed 2-propanol/hexane (55:45, *v*/*v*) in capped screw-top tubes stored at −20 °C until analyzed. Cholesterol was extracted from the egg yolks via saponification with potassium hydroxide followed by petroleum ether extraction according to the method of Beyer et al. [25]. Briefly, a 0.25 mL sample of Folch total lipid extract was vortexed with 2.5 mL of freshly prepared 2% (*w*/*v*) KOH in absolute ethanol and incubated at 37 °C for 1.5 h. Then, 5.0 mL of petroleum ether was added, followed by 2.5 mL of distilled water. The upper layer was quickly placed in a tightly capped tube to prevent evaporation of the ether and, thus, the concentration of the sample and then quantified via HPLC (Thermo Scientific™ Vanquish™, Thermo Scientific, Waltham, MA, USA) with UV detection (wavelength of 210 nm), which was equipped with a Kinetex 5 μL C18 reverse-phase column (150 cm × 4.6 mm × 5 μm; Phenomenex, Torrance, CA, USA). The HPLC mobile phase consisted of acetonitrile/2-propanol (55:45, *v*/*v*) at a flow rate of 1.0 mL/min. The injection volume for all samples was 10 µL. Identification and quantification of the peaks were compared with cholesterol standards (30–500 µg/mL). Results are expressed as mg/g fresh matter for total cholesterol.

### 2.4. Fatty Acid Composition

The fatty acid composition of the egg yolks was quantified as methyl esters (FAME) according to the method described by Stoffel et al. [26] using a gas chromatograph GC Trace 2000 (ThermoQuest EC Instruments, Milan, Italy) equipped with a flame ionization detector (260 °C) and a fused silica capillary column (Zebron ZB-88, Phenomenex, Torrance, CA, USA) of 100 m × 0.25 mm × 0.25 μm film thickness. Helium was used as a carrier gas. The column temperature was held at 100 °C for 5 min and then raised by 4 °C/min up to 240 °C and maintained for 20 min. The individual fatty acid peaks were identified by comparing retention times with those of known mixtures of standard fatty acids (PUFA 2, Animal Source Supelco, Bellefonte, PA, USA) run under the same operating conditions. Results are expressed as percentages of the total FA identified. To assess the nutritional value of the intramuscular fat, the following indexes were calculated: the n-6 PUFA to n-3 PUFA ratio (n-6/n-3); the PUFA to SFA ratio (PUFA/SFA); and the atherogenic index (AI) and the thrombogenic index (TI) to evaluate the risk of atherosclerosis and the potential aggregation of blood platelets, calculated according to the formulas suggested by Ulbricht and Southgate [27].

### 2.5. Statistical Analysis

Data were analyzed using the GLM procedure of SPSS (IBM SPSS Statistic Data Editor version 25, Chicago, IL, USA) considering the production system and hen age as the main factors. Significant differences between means were verified with a post hoc Scheffe’s test (*p* < 0.05). Mean values and standard errors of the mean are reported.

To simplify the difference in fatty acid profiles between different groups, we utilized a principal component analysis approach (PCA; Johnson and Wichern, 1992) to reduce the dimensionality of the problem by replacing these fatty acids components with the first two PCAs, explaining about 90% of the total variability in these variables (PCA results were from the analysis function of “princomp” and were plotted by the “factoextra” package, implemented in R version 4.3.2).

## 3. Results and Discussion

### 3.1. Total Lipid and Cholesterol Content

Most lipids are contained in the egg yolk, while a modest amount is associated with the vitelline membranes. Egg yolk contains a complex mixture of lipids mainly consisting of triglycerides (62%), phospholipids (33%), and cholesterol (5%) [28]. The lipid composition of egg yolk is a primary consumer concern especially in relation to cholesterol content, as it was viewed in the past as a negative nutrient. However, recent evidence from randomized controlled trials suggests that eggs tend to have overall small effects on blood cholesterol levels [5]. Previously, Qureshi et al. [29] demonstrated that consuming more than six eggs per week or one egg or greater per day did not increase the risk of coronary artery disease. Therefore, the dietary recommendations restricting egg consumption should be taken with caution and should not include all individuals; we need to acknowledge the multiple beneficial effects of the inclusion of eggs in regular diets as well. The cholesterol concentration in eggs and yolk can be influenced by different factors (breed; age of layers; management and nutritional strategies) [3,30]. Based on the above, various nutritional attempts have been conducted to reduce the content of cholesterol and modify the fatty acid profile to obtain n-3 PUFA-enriched eggs [31,32,33,34,35]. In the present study, we found a total lipid content of 33%, similar to those reported in other studies on different genotypes of laying hens [7,36]. The cholesterol content found in the present work (9.7 mg/g of yolk) was similar to the values found by Racevičiūtė-Stupelienė et al. [7] on Lohmann Brown Classic line laying hens, ranging from 8.43 to 8.96 mg/g yolk. Conversely, quite high values were found by Tomaszewska et al. [31] in Bovans Brown laying hens at 60 weeks (11.73 ± 0.17 mg/g yolk) and by Zemkovà et al. [37] in ISA brown laying hens (ranging from 11.2 to 16.1 mg/g yolk). 

Regarding the effect of production systems and hen age, neither the total lipids (ranging from 32.04% to 33.45%) nor cholesterol (ranging from 9.20 mg/g to 9.98 mg/g) of the egg yolks (Table 2) were affected by production systems or hen age (*p* > 0.05). Similarly, Sokołowicz et al. [38] did not find any significant effect of the production system on egg yolk cholesterol content in native Greenleg Partridge hens (Z-11), Rhode Island Red, or Hy-line Brown commercial hens from litter barn, free-range, and organic systems. Racevičiūtė-Stupelienė et al. [7] did not find any effect of the production system (cage and barn) or age (28, 38, and 48 weeks of age) on cholesterol content in Lohmann Brown Classic line-laying hens. Conversely, Zemková et al. [37] found that the housing system and the age of hens had a significant effect on both yolk and egg cholesterol concentrations, which were lower in eggs from enriched cages compared with those from a litter system; age had a fluctuating effect on cholesterol content, which decreased from 50 to 59 weeks followed by an increase at 68 weeks and, thereafter (at 75 weeks), a decrease. 

### 3.2. Fatty Acid Composition

Table 3 presents the effects of the production systems and hen age on the total fatty acid composition of egg yolk. Overall, total MUFA content was predominant, followed by SFA and then PUFA; a higher proportion of total n-6 PUFA (+16.8%) compared with total n-3 PUFA was also found. This distribution of the proportion of fatty acids in yolk agrees with results reported on different genotypes of laying hens [3,39,40,41]. Conversely, Racevičiūtė-Stupelienė et al. [7] found a higher value of SFA, lower PUFA, and an intermediate value of MUFA in the egg yolks of Lohmann Brown Classic hens. These differences could be due to the fatty acid composition of the feed [3]. The predominant FAs in yolk egg in all groups were palmitic acid (C16:0) such as SFA, oleic acids (C18:1) such as MUFA, and linoleic acids (C18:2 n-6, LA) such as PUFA. Oleic acid (ranging from 39.64% to 45.60%) was the highest FA, in accordance with the results of recent research [3,40].

Except for the total SFA content, total MUFA, PUFA, n-3 PUFA, and n-6 PUFA contents were affected by the production systems (*p* < 0.01). In particular, yolks from the free-range system had higher n-3 PUFA and n-6 PUFA content, showing major differences (*p* < 0.01) in linoleic acid (C18:2 n-6, LA), and lower MUFA, mainly due to the lower proportions (*p* < 0.01) of oleic acid (C18:1) and palmitoleic acid (C16:1); the contribution from other individual acids was very low due to their very small amounts (Table 4). This finding was partially supported by the results reported by previous works [3,7,42]. As extensively reported in the literature, the results of the present study confirmed that the PUFA n-3 content of eggs can be simply modified by diet: eggs from the free-range system showed almost twice the content recorded for cage eggs. 

Total FA (SFA, MUFA, PUFA, n-3 PUFA, and n-6 PUFA) and individual fatty acids did not differ significantly from the 68th to 74th week of age (*p* > 0.05). In a recent study carried out by Zita et al. [43] on eggs from Lohmann Brown layers—analyzed at the beginning of the first (from the 28th to 30th week of age) and second (from the 78th to 80th week of age) laying cycle—it was found that egg yolks from the second laying cycle had higher MUFA and lower SFA and PUFA contents compared with those of the first laying cycle. In this case, the highest value of MUFA in the second laying cycle was due to a higher amount of oleic acid (+6%), probably due to the significant impact of feeding management on oleic acid [44] or different total lipid contents in the egg yolk of the second laying cycle. Ko et al. [45]—in a study carried out on eggs from Lohmann Lite Brown laying hens of early (24 weeks), intermediate (42 weeks), and late age (74 weeks)—found that egg yolk from older hens had lower SFA and MUFA content compared with intermediate- and early-age laying hens but similar PUFA content compared with intermediate-age hens. The age-related differences found by the above-mentioned studies could be related to differences in hen metabolism, which is in function of the animal’s age and influences changes in the yolk’s fatty acid composition [46]. In addition, it is known that the effect of hen age is tied to breed; diet and feed composition; and environmental conditions [44]. 

The fatty acid profiles of the egg yolk samples were recorded with principal component analysis (PCA). It should be noted that the different age groups completely overlap, whereas the C group and FR group only slightly overlap (Figure 1). The fatty acid profiles of egg yolks from different age groups were similar but apparently not between different production systems.

The effects of the production systems and hen age on the single fatty acids of the egg yolks are reported in Table 4.

The FA profile of egg yolks in laying hens is strongly dependent on the FA profile of the diet [3,41,47], which has been confirmed in the present study. In fact, free-range yolks can be distinguished from those of the cage system. These differences could be due to the synergistic combination of the different fatty acid compositions of the diets and the availability of grazing and, therefore, the intake of different grass essences. Partially agreeing with our results, Anderson [42] found that eggs from the range production system had higher total fat, MUFA, and PUFA than eggs produced by caged hens, which can be reasonably explained by forage consumption and the intake of wild seeds or insects associated with pastures from free-range areas, as it has been reported that there are species of edible insects with fat levels as high as 31.4% [48]. From a nutritional point of view, individual n-3 fatty acids, such as α-linolenic acid (ALA, C18:3n3), eicosapentaenoic acid (EPA, C20:5n3), docosapentaenoic acid (DPA, C22:5n-3), and docosahexaenoic acid (DHA, C22:6n3), were higher in the FR system compared with the cage system (*p* < 0.05 and *p* < 0.01). Based on current cardiovascular evidence, the day-to-day consumption of a healthy amount of essential fatty acids for the adult population should be 250 mg of eicosapentaenoic acid (EPA) plus docosahexaenoic acid (DHA) [49]. If we take into account the average lipid content observed in the yolks of the present study, which was about 32.9 g/100 g, and the average content of EPA + DHA, 0.48%, the intake of these long-chain n-3 PUFAs per day (157.9 mg/100 g) is able to satisfy about 63.2% of daily long-chain n-3 PUFA requirements. Compared with the C group, FR showed a numerically higher incidence of EPA + DHA calculated for total lipid content (47.7 vs. 110.2 mg/100 g, respectively).

### 3.3. Nutritional Indices

The results presented in Table 5 show that the nutritional indices—such as n-6/n-3 FAs, PUFA/SFA, the atherogenic index (AI), and the thrombogenic index (TI)—of the egg yolks were not affected by the increasing age of the layer from the 68th week to the 74th week, while the difference was significant between eggs from different production systems. 

For years, emphasis has been placed on the potential pro-inflammatory properties of some n-6 PUFA metabolites (e.g., some of the eicosanoids deriving from arachidonic acid) [50] and on the competition between LA and ALA as substrates for the same metabolic pathway, possibly leading to reduced levels of ALA-derived n-3 long chain PUFA (LC-PUFA) in organs and tissues [51]. Obviously, in our case, the lower ratio of n-6/n-3 FAs in egg yolks from the FR system is favorable. Conversely, Dalle Zotte et al. [3]—in a study carried out on marketed table eggs derived from different farming methods (cages, barns, and organic) and sampled in different marketing periods—found a higher n-6/n-3 ratio in eggs from an organic system compared with a cage one (19.2 vs. 15.4). Regarding the age-related effect, it was found that, in eggs from late-age laying hens, it was higher compared with early-age laying hens (22.3 vs. 18.2) but similar to intermediate ones (22.2) [45]. Conversely, Zita et al. [43] did not find any age-related effects on the n-6/n-3 ratio during the first and second laying cycles (16.56 vs. 18.24, respectively). PUFA/SFA is a commonly used index for evaluating the nutritional value of dietary foods; it hypothesizes that all PUFAs in the diet can depress low-density lipoprotein cholesterol (LDL-C) and lower levels of serum cholesterol, whereas all SFAs contribute to high levels of serum cholesterol. Thus, the higher this ratio, the more positive the effect [52]. The PUFA/SFA ratio presented in our study is in the context of it as is well known being suggested to be above 0.4 in the diet, the FR system can produce a favorably higher PUFA/SFA ratio than the cage system (*p* < 0.01). The AI and TI were introduced as measures of the connection of the diet to the incidence of coronary heart disease [27], and lower AI and TI values have been recommended for a healthy diet [53]. The production systems did not affect the AI, but the TI was significantly and favorably lower in egg yolks from the FR system. Dalle Zotte et al. [3] did not find any significant effect of production systems (cage, barn, and organic) on the above-mentioned indexes. Regarding the age-related effect, it was found that the fatty acid composition of egg yolks from late-age laying hens resulted in lower AI and TI values compared with early-age laying hens [43,45], and the detected values were in the same range of our results.

In the present study, some significant interactions were found between the two factors studied; the main ones regarded the total n-3 and n-6 PUFAs (*p* < 0.01), the n-6/n-3 ratio (*p* < 0.01), and the TI index (*p* < 0.05). In particular, it was observed that, in the cage farming system, the n-6/n-3 ratio was markedly reduced in the yolks of eggs produced at 74 weeks of age (Figure 2A) due to a reduction in the total n-6 PUFA content (FR 74 d = 22.36%; C 74 d = 11.08%). Conversely, the TI index (Figure 2B), was higher in the egg yolks of 74-week hens from the C group (FR, 74 weeks = 1.13; C, 74 weeks = 1.00).

## 4. Conclusions

The egg yolk lipid profile, in terms of the total lipid, cholesterol, and fatty acid composition, was not modified by the increasing age of laying hens between the 68th and 74th week. This information could be relevant for farmers and consumers since there is a growing trend in extending the laying cycle to improve the economic and environmental sustainability of the egg production system. While this practice is biologically feasible and beneficial to the environment, it comes with some challenges, such as maintaining hen health and laying persistence while also maintaining the quality of external and internal eggs. All these aspects require in-depth knowledge of the physiology of hens and their nutritional needs, which can be guaranteed by adequate support to the farmer from feed companies and by collaboration with research centers such as universities. Our results confirmed that the free-range system may improve the nutritional yolk fatty acid profile and its positive impact on human health, which can be a convincing support for the trend of “cage-free” eggs. However, considering this is the first study to compare the lipid profiles of egg yolks from the 68th to 74th weeks of laying, more studies are needed to gain a more comprehensive understanding of the advantages of “cage-free” eggs throughout the studied period of laying.

## Figures and Tables

**Figure 1 animals-14-01099-f001:**
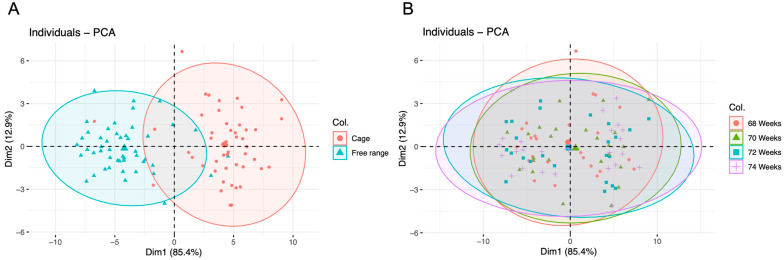
Principal component analysis of fatty acid profiles of different production systems (**A**) and different age (**B**) projections of the variables in the plane defined by the two principal components. Cage: enriched cage system; free range: free-range system; 68 Weeks: 68th week of age; 70 Weeks: 70th week of age; 72 Weeks: 72nd week of age; 74 Weeks: 74th week of age.

**Figure 2 animals-14-01099-f002:**
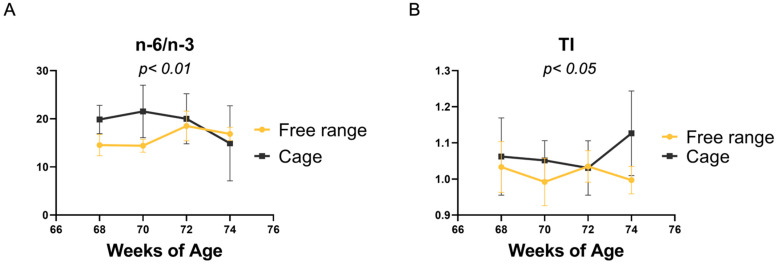
Interactions between production systems and hen age in the PUFA n-6/n-3 ratio (**A**) and thrombogenic index (**B**) of egg yolks.

**Table 1 animals-14-01099-t001:** Chemical nutritional characteristics of the diets.

Item * (%)	Enriched Cages (C)	Free Range (FR)
Dry matter	88.85	89.59
Crude protein	15.54	16.91
Ash	13.30	11.99
Crude fat	4.12	4.06
Fiber	5.53	5.92
Lysine	0.81	0.83
Methionine	0.43	0.43
Calcium	4.00	3.45
Sodium	0.15	0.17
Phosphorus	0.53	0.59
Fatty acid profile (% of total fatty acids)		
C14:0	0.38	0.31
C16:0	18.83	14.09
C18:0	6.16	2.26
SFA	25.38	16.66
C16:1n9	0.35	0.33
C18:1n9	29.62	26.86
C20:1n9	0.10	0.03
MUFA	30.07	27.21
C18:2n6	41.27	51.93
C20:4n6	0.17	0.14
C22:4n6	0.05	0.12
C20:5n3	0.14	0.25
C18:3n3	2.82	3.56
C22:5n3	0.06	0.06
C22:6n3	0.04	0.21
n-6 PUFA	41.49	52.16
n-3 PUFA	3.06	3.97
PUFA	44.55	56.13

* Ingredients: GMO-free corn, sunflower meal, calcium carbonate, GMO-free soybean meal, dried distillers’ grains without GMO, raw sunflower oil, linseed, sodium chloride, monocalcium phosphate, and sodium sulfate. The producer reserves detailed information on each ingredient (Tasomix, Biskupice Ołoboczne, Poland). Provided per kilogram of diet: vitamin A (retinyl acetate), 10,000 IU; vitamin D3 (cholecalciferol), 3000 IU; vitamin E (DL-α_tocopheryl acetate), 20 IU (C) and 10 IU (FR); Fe, 50 mg; Cu, 8.0 mg; Zn, 60 mg; Se, 0.15 mg; Mn, 70 mg; I, 0.9 mg.

**Table 2 animals-14-01099-t002:** Effect of production systems and hen age on total lipid and cholesterol content of egg yolk.

	Production System (PS)		Hen Age, Weeks (A)				SEM	Significance		
	Cage	Free Range	68	70	72	74		PS	A	PS × A
Cholesterol, mg/g	9.90	9.59	9.20	9.86	9.94	9.98	0.137	ns	ns	ns
Lipids, %	32.75	33.00	33.32	32.04	32.70	33.45	0.265	ns	ns	ns

SEM = standard error mean; ns = not significant.

**Table 3 animals-14-01099-t003:** Effect of production systems and hen age on total fatty acid composition (% of total fatty acids) of egg yolk.

	Production System (PS)		Hen Age, Weeks (A)				SEM	Significance		
	Cage	Free Range	68	70	72	74		PS	A	PS × A
ƩSFA	35.99	35.79	36.14	35.69	35.93	35.80	0.152	ns	ns	ns
ƩMUFA	47.96	41.64	44.33	45.38	44.77	44.72	0.216	**	ns	ns
ƩPUFA	16.05	22.57	19.53	18.93	19.29	19.48	0.212	**	ns	ns
Ʃn-6 PUFA	14.84	20.95	18.05	17.99	18.80	16.72	0.294	**	ns	**
Ʃn-3 PUFA	0.79	1.33	1.11	1.08	1.00	1.04	0.017	**	ns	**

SFA = saturated fatty acid; MUFA = monounsaturated fatty acid; PUFA = polyunsaturated fatty acid. SEM = standard error mean; ns = not significant; ** *p* < 0.01.

**Table 4 animals-14-01099-t004:** Effect of production systems and hen age on single fatty acids (% of total fatty acids) of egg yolks.

	Production System (PS)		Hen Age, Weeks (A)				SEM	Significance		
	Cage	Free Range	68	70	72	74		PS	A	PS × A
C14:0	0.42	0.36	0.40	0.39	0.39	0.39	0.006	**	ns	ns
C16:0	28.62	28.40	28.78	28.35	28.55	28.37	0.175	ns	ns	ns
C18:0	6.94	7.03	6.97	6.94	6.99	7.04	0.055	ns	ns	ns
C16:1	2.28	1.87	2.20	2.08	2.07	1.95	0.053	**	ns	ns
C18:1	45.60	39.64	42.02	43.20	42.59	42.66	0.221	**	ns	ns
C20:1	0.09	0.13	0.11	0.11	0.11	0.10	0.107	**	ns	ns
C18:2 n-6	13.62	19.72	16.85	16.27	16.71	16.85	0.200	**	ns	ns
C18:3 n-3	0.45	0.61	0.57	0.53	0.50	0.51	0.009	**	ns	**
C18:3 n-6	0.04	0.05	0.04	0.04	0.05	0.04	0.001	ns	ns	ns
C20:3 n-6	0.08	0.08	0.08	0.08	0.08	0.08	0.001	ns	ns	ns
C20:4 n-6	1.42	1.32	1.36	1.37	1.38	1.38	0.017	**	ns	ns
C20:5 n-3	0.02	0.03	0.02	0.03	0.03	0.02	0.001	*	ns	ns
C22:4 n-6	0.09	0.07	0.08	0.08	0.09	0.08	0.002	**	ns	ns
C22:5 n-3	0.04	0.05	0.05	0.05	0.04	0.05	0.002	**	ns	ns
C22:6 n-3	0.27	0.65	0.48	0.48	0.43	0.46	0.010	**	ns	**

SEM = standard error mean; ns = not significant. * *p* < 0.05; ** *p* < 0.01.

**Table 5 animals-14-01099-t005:** Effect of production systems and hen age on nutritional indices of egg yolk.

	Production System (PS)		Hen Age, Weeks (A)				SEM	Significance		
	Cage	Free Range	68	70	72	74		PS	A	PS × A
n-6/n-3	19.08	16.08	17.20	17.97	19.27	15.88	0.434	**	ns	**
PUFA/SFA	0.45	0.63	0.54	0.53	0.54	0.55	0.007	**	ns	ns
AI	0.48	0.47	0.48	0.47	0.47	0.48	0.004	ns	ns	ns
TI	1.07	1.01	1.05	1.02	1.03	1.06	0.008	**	ns	*

AI = atherogenic index; TI = thrombogenic index; SEM = standard error mean; ns = not significant. * *p* < 0.05; ** *p* < 0.01.

## Data Availability

The data presented in this study are available upon request from the corresponding authors. The data are not publicly available due to restrictions imposed by the research group.

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
