# Peer review of "Comparison of Fatty Acid Profile in Egg Yolk from Late-Age Hens Housed in Enriched Cages and in a Free Range System"

_animals, 2024, doi:10.3390/ani14071099_

Round 1

Reviewer 1 Report

Comments and Suggestions for Authors

Manuscript is written in very high scientific as well as professional level. Paper brings many additional and/or original findings in poultry science area.

However, my opinion is that MINOR revision is needed:

1. in whole paper I recommend to use term crude fat (instead of total lipid), if the indicator is after extraction, f.e. via Soxhlet methodology.

2. biological material was from practise, from commercial farm in Poland. However, in similar papers, information about local commitee agreement is usual. so, if possible, add this information to the text.

3. in methodology, deeper information about the feed mixture preparing is needed. can You add simply information about the technology of mixture producing?

After the revision, I recommend this paper for publishing in journal Animals.

Reviewer 2 Report

Comments and Suggestions for Authors

Probably the topic of evaluated paper is not very innovative but its general quality is quite high. Nevertheless, several points need clarification.

It seems that the study material was standardised, same hybrid birds, same age etc. But why was a different feed mixture used? This mixture itself contains more unsaturated fatty acids, which certainly affected the fatty acid profile of the yolks. So feeding and housing were not separated, apart from access to the run, also the feed should be considered as an experimental factor.

There is also a lack of information on what the runs looked like. Were they covered with green vegetation? If so, what kind of plants? How much time did the birds spend in the open-air run? Was it only fenced or also roofed (due to epidemic requirements)?

Figure 2 is redundant due to it is a repetition of data from table 5.

Reviewer 3 Report

Comments and Suggestions for Authors

Dear Authors,

Information about the importance of cholesterol and fatty acids should be given in the introduction.

A clear sentence must be written about why the study is needed.

Line 43: Replace “ farming” with “production”

Line 54-63: This part was written very long. You need to summarize this part a little more.

Line 70: Replace “ rearing” with “production”

Line 157: Please specify how atherogenic and thrombogenic indexes are calculated. Why are these features needed? Express the importance of this in one or two sentences.

Table 3: You need to indicate the differences between groups by lettering.

Table 4: You need to indicate the differences between groups by lettering.

Line 70: Replace “ rearing” with “production”, and throughout the article.

Reviewer 4 Report

Comments and Suggestions for Authors

Studies involving the effects of laying hen housing systems on egg quality are necessary and welcome. However, the manuscript needs adjustments so that it is ready to be published.

Line 12. Replace the word 'rewarding' with 'necessary'.

Introduction - Please improve the state of the art to clearly show progress on the topic covered in the study. Includes more information about the lipid profile of eggs in the yolk and its importance for human health. When reading the introduction it appears that the main focus of the article is bird welfare. The lack of adequate justification for carrying out the study creates the wrong impression that the authors are unaware of recent developments. Use references that demonstrate that there is a need for new experiments involving the laying hen housing system (DOI: 10.1016/j.livsci.2021.104597). Please use relevant recent references that address the nutritional profile of eggs. This would give readers a sense of continuity and help them place your article in context, greatly strengthening the impact of your article.

Row 48-49. Delete the sentence. The description of the conventional system can generate mistaken and biased interpretations.

Material and methods

In general, the nutritional profile of eggs is quite constant unless there is some intervention in the birds' diet. In table 1 we note the difference in diets for each housing system. In this case, it is difficult to say that the effect found is a response from the housing system. Why are the diets so diferente (Difference in fatty acid profile)?

Add information about the forage consumption of free-ranging birds. This may also contribute to the observed difference in results.

Do you believe that the number of eggs used in the experiment (from just one producer) is sufficient?

Discussion

Line 240. It would be interesting to explain why this occurs.

Is there any relationship between the lipid profile of eggs and the age of chickens?

It would be interesting to include some biological/physiological explanation. This information will support the discussion and mainly demonstrate the importance of the study. We know that the problem for birds at the end of the production cycle is the reduction in eggshell quality. In this sense, bring more information to support the need and motivation for carrying out your study.

Line 265. Not a hypothesis. It's a fact!

Reviewer 5 Report

Comments and Suggestions for Authors

Minor edits shown in attached file

Two references that pertain directly to this research are relevant to your conclusions: Anderson 2013 and 2011 in Poultry Sci 92:3259 and 90:1600

Comments on the Quality of English Language
